# CHARM: Creating Halos with Auto-Regressive Multi-stage networks

**Shivam Pandey**
Columbia Astrophysics Laboratory, Columbia University, NY

**Chirag Modi**
Center for Computational Astrophysics, Flatiron Institute, NY
Center for Computational Mathematics, Flatiron Institute, NY

**Benjamin Wandelt**
CNRS & Sorbonne Université, Institut d'Astrophysique de Paris (IAP), Paris, France
Center for Computational Astrophysics, Flatiron Institute, NY

**Guilhem Lavaux**
CNRS & Sorbonne Université, Institut d'Astrophysique de Paris (IAP), Paris, France

## Abstract

To maximize the amount of information extracted from cosmological datasets, simulations that accurately represent these observations are necessary. However, traditional simulations that evolve particles under gravity by estimating particle-particle interactions (N-body simulations) are computationally expensive and prohibitive to scale to the large volumes and resolutions necessary for the upcoming datasets. Moreover, modeling the distribution of galaxies typically involves identifying collapsed and bound dark matter structures called halos. This is also a time-consuming process for large N-body simulations, further exacerbating the computational cost. In this study, we introduce CHARM, a novel method for creating mock halo catalogs by matching the spatial and mass statistics of halos directly from the large-scale distribution of dark matter density field. We develop multi-stage neural spline flow based networks to learn this mapping directly with computationally cheaper, approximate dark matter simulations instead of relying on the full N-body simulations. We validate that the mock halo catalogs have same statistical properties as obtained from traditional methods. Our method effectively provides a speed-up of more than a factor of 1000 in creating reliable mock halo catalogs compared to conventional approaches. This study represents a major first step towards being able to analyze the non-Gaussian and non-linear information from current-generation surveys using simulation-based inference approaches on the massive scales of upcoming surveys.

## 1 Introduction

The standard model of cosmology describes the evolution of the Universe using a set of free cosmological parameters. Constraining these parameters with observations is the primary goal of cosmology. Over approximately 13.7 billion years of evolution, the hierarchical structure formation process transforms the initial Gaussian distribution of matter into a highly non-Gaussian field comprising of halos, voids, and filaments. The observed galaxies occupy the collapsed and bound structure of

NeurIPS 2023 AI for Science Workshop.

dark matter called halos. The cosmological parameters can be constrained by analyzing the statistical distribution of the observed galaxies and comparing them to predictions from theoretical models or simulations. Traditional techniques limit this comparison to simple two-point summary statistics, such as the power spectrum at large scales, as theoretical models break down for higher-order statistics and non-linear small scales [18, 4, 3]. Since the evolved matter distribution is non-Gaussian, higher-order statistics, as well as small-scale two-point correlations, carry a significant amount of complementary information about the cosmological parameters [9, 14, 1, 6, 21, 17].

To extract this information, we need to rely on accurate N-body simulations and employ simulation-based inference (SBI) techniques [9, 10]. SBI involves using computational forward models to simulate the data on a grid of cosmological parameters, measuring the statistics of interest and comparing it with the observed data using machine learning techniques to constrain the parameters. For galaxy clustering surveys, these forward models involve evolving the dark matter particles under gravity, identifying the dark matter halos in each scenario, and then populating these halos with galaxies. However, simulating the high-mass halos (which are very rare) in various environmental conditions requires a large simulation box, while simulating the lower halo masses demands high resolution. Particularly, to reliably analyze the current generation of galaxy surveys, the number of particles and the volume of the simulation required are so large that the computational cost of running these simulations at a grid of cosmological parameters is prohibitive. To put things in context, to analyze the last generation of cosmological surveys which ended a decade ago with this approach would require running at least 2000 simulations with $2.7 \times 10^{10}$ particles, taking more than 270 million CPU hours for running the simulations alone [22]. Furthermore, finding the halos in these N-body simulations also adds to the computational cost.

However, physically, we expect the number and mass distribution of halos to depend on the large-scale matter distribution. For example, the overdense regions of the universe will have more matter to collapse and will be able to form more numerous and heavier halos. Therefore, accurately learning this relationship and generating fast approximations to the dark matter distribution on large scales can accelerate mock halo catalog generation, and ultimately generating observed data with end-to-end simulations. This motivates us to use deep learning techniques to learn these highly non-linear and non-local relationship between the dark matter and halo distribution.

In order to further accelerate the simulations on large scales, particle mesh (PM) approximations can be used [e.g., 19, 7]. These approximations estimate the gravitational forces by interpolating CDM (cold dark matter) particles on a uniform grid, enabling the use of techniques such as fast Fourier transforms to solve the equations of motion. Due to this grid interpolation, they lose information on scales smaller than the grid resolution, resulting in poor halo catalogs. However, on scales larger than the grid size, they accurately capture the matter distribution. Since these PM simulations are orders of magnitude faster than N-body simulations, our goal is to learn the relationship between halos and matter density obtained from PM simulations, instead of the N-body simulations.

In this work, we introduce CHARM: a generative model for creating halo catalogs using multi-stage neural spline flows from transforming the low-resolution PM simulations to discrete mock catalog expected from a high-resolution N-body simulation. It consists of 3 stages after extracting features extracted from the surrounding dark matter density in PM simulations at any location: $(i)$ learn the number of halos expected, $(ii)$ learn the mass of the heaviest halo and, $(iii)$ auto-regressively predict lower halo masses. Developing a methodology like this is crucial for using simulation-based inference techniques to analyze and maximizing the information gain from current and future surveys.

## 2 Related Work

In recent years, there have been other studies with related goals, but they provide different solutions compared to what is desired here. In [2], a similar mapping is learned using physically motivated networks, but they assume an explicit form of likelihood for halo occupation, which breaks down for high-mass halos and small scales, which are of interest in this study. There have also been attempts, as in [15] and [23], that do not impose a likelihood form. However, the methodology of [15] only works for continuous fields like total halo mass, whereas here we aim to obtain discrete halo catalogs. The methodology of [23] is designed to work only with dark matter density obtained from high-resolution N-body simulations, thus requiring large computational resources.

In [12], displacement corrections to the PM simulations were provided to make them resemble their N-body counterparts. However, these corrections do not extend to very small scales, and hence the recovered halo catalogs are not fully accurate. Nevertheless, these corrections could be used to augment the PM simulations used here and improve the accuracy of the model in the future.

## 3    Dataset

**Simulations:**    We use the public simulation suite from the N-body Quijote project [22], which simulates a volume of approximately $1000(\mathrm{Mpc}/h)^3$, where Mpc is one mega-parsec (approximately $3 \times 10^6$ light years), and $h$ is the dimensionless Hubble parameter that is proportional to the expansion rate of the Universe. These simulations have enough volume and resolution to provide reliable dark matter halo catalogs with masses above $10^{13} M_\odot/h$, where $1 M_\odot$ is equal to the solar mass. Therefore, these simulations provide a good suite to build a reliable model of how the distribution of halos relates to the dark matter field around them, which can then be applied to significantly larger volume simulations. In this first work focused on that goal, we fix the cosmological parameters and process 20 independent simulations with different initial conditions. Each N-body simulation evolves an initial Gaussian distribution of $1024^3$ cold dark matter particles to the present time and takes approximately 5000 CPU hours to finish.

**Input dark matter density:**    For the approximate simulations, which will form our input, we run 20 paired simulations (i.e., matching Gaussian initial conditions and cosmology) using the FASTPM algorithm [7]. Here, we evolve only $256^3$ particles over the same volume, and each simulation takes only 3 CPU hours[1]. We compute matter density fields ($\rho_m$) from the PM simulations on a regular grid with $128^3$ voxels using cloud-in-cell interpolation. We convert this density field to the over-density field ($\delta_m = \rho_m/\bar{\rho}_m - 1$), where $\bar{\rho}_m$ is the average matter density of each simulation box. Note that this means that each voxel has a physical box length of approximately 7.8 Mpc/$h$. Additionally, it's important to note that the size of a typical halo is less than 1 Mpc/$h$, and hence each voxel can host multiple halos.

**Target halo catalog:**    From the N-body simulations, we voxelize our target halo distribution on the same $128^3$ grid using the nearest-grid point mass assignment scheme. Within this scheme, for each voxel $i$, we count the number of halos inside it ($N_{\mathrm{tot}}^i$), and if it is non-zero, we also store the halo masses in decreasing order ($[M_1^i, M_2^i, \ldots, M_{N_{\mathrm{tot}}^i}^i]$, where $M_1^i > M_2^i > \ldots > M_{N_{\mathrm{tot}}^i}^i$). In our training set, we have at most six halos in any voxel, i.e., $N_{\max} = 6$, and $N_{\mathrm{tot}}^i \leq N_{\max}$ for all $i$. Moreover, we impose a minimum halo mass cut of $M_{N_{\mathrm{tot}}^i}^i \geq 10^{13} M_\odot/h$ for all $i$, which is the relevant halo mass range for current-generation galaxy surveys.

**Augmentations:**    We select 10 of the paired simulations to create the training dataset. We divide the 3D simulation boxes, each of size $128^3$, into sub-boxes of size $16^3$, resulting in 512 sub-boxes from each of the simulations. Each sub-box has a physical size of 125 Mpc/$h$. Next, in our training set, we sort the $512 \times 10$ sub-boxes from the N-body simulations in decreasing order based on the mass of their heaviest halos and select the first 512 sub-boxes from this sorted list. The reason for doing this is that these heaviest halos are exponentially rare, but host a significant fraction of the galaxies, hence over-representing them in the training dataset leads to better results. We choose the same sub-boxes from the PM set to create the input dark matter density field, paired with the N-body halos for training. To facilitate feature extraction, we pad the input density field from the PM sub-boxes such that the output after convolutions preserves the size of sub-box ($16^3$). Therefore, we add a padding of four voxels on each side from the original periodic simulations.

## 4    Methodology

The task of this paper is to obtain a discrete mock halo catalog when provided with an approximate dark matter over-density field from PM simulations. Halo formation is a complex non-linear process that depends on the 3D matter distribution on large scales. For example, there is a higher probability of forming a heavy halo at the intersection of large dark matter filaments compared to in a void region.

---

[1]GPU implementations of these algorithms can further increase computational efficiency [16, 13]

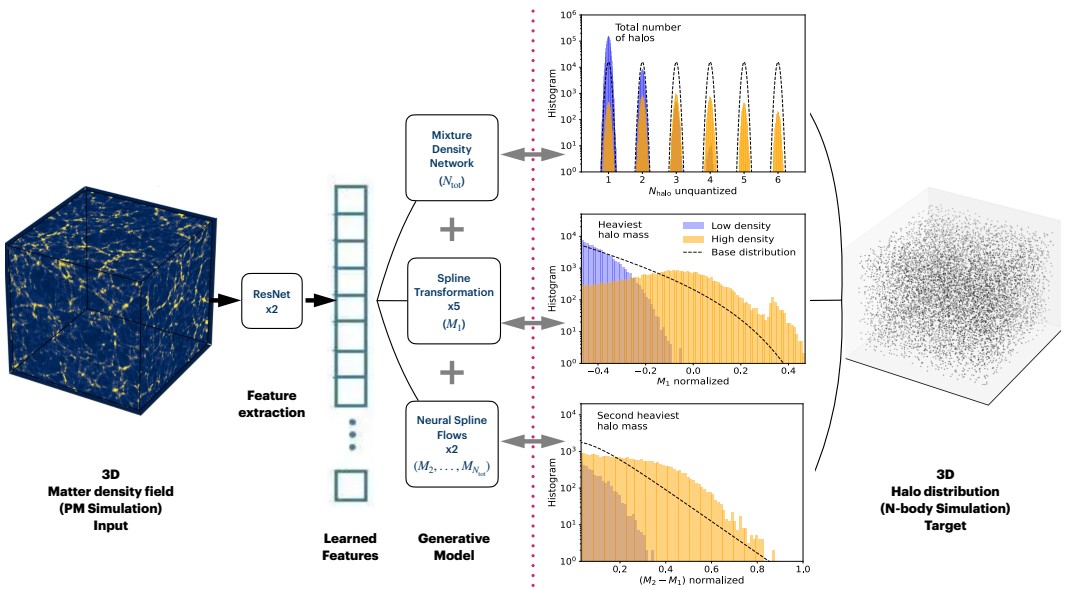

Figure 1: Visualization of the data products and network architecture used in this study. On the left side of the dashed line, we show the matter density field from PM simulation that serves as the input to the ResNet layers to extract features. These features are then used to predict the halo distribution in three parts: the total number of halos is modeled using a mixture density network, the heaviest halo mass is modeled using a stack of spline transformations, and lower halo masses are modeled using a stack of auto-regressive neural spline flows. On the right side, we display the target distribution of the halos from paired N-body simulation. Additionally, we present histograms of the three quantities in both low and high-density voxels, illustrating their dependence on the dark matter distribution.

Therefore, to extract the features of dark matter density that correlate with the halo distribution, we stack two 3D residual network (ResNet) layers [11]. These features, extracted from a physical region of approximately 70 Mpc/$h$, are used as conditioning for a multi-level generative model for the halo distribution, as described below.

To create a mock halo catalog, we need to estimate a discrete distribution of halos conditioned on a feature vector for each voxel. To achieve this, we split the problem into three parts. For each voxel $i$, we first predict the total number of halos ($N_{\text{tot}}^i$), which provides a mask as well as an occupation number to train the mass distribution. Then, for the voxels that have a non-zero number of halos, we predict the mass of the heaviest halo ($M_1^i$). Finally, for voxels that have more than one halo, we train the prediction for the masses of lighter halos ($M_2^i, \ldots, M_{N_{\text{tot}}^i}^i$) in an auto-regressive fashion. This means that, as dictated by the physics of structure formation, we always condition the probability of lighter halo masses on the masses of all the heavier halos in the same voxel. Our final loss function is a sum of the losses from these three steps and the details of each step is as follows (also see Fig. 1 for a brief summary of this inference pipeline):

1. We model the probability distribution of the total number of halos as a mixture of $N_{\max}$ Gaussians. We take the discrete distribution of the number of halos in each voxel of the training simulations and make it continuous by adding a small Gaussian noise with a known variance. With input from the learned features of the ResNet, we predict the probability of each Gaussian using a fully connected neural network (FCNN). In this case, the loss is modeled as the forward Kullback–Leibler (KL) divergence between the modeled Gaussian mixture with predicted probabilities (with fixed mean and variance) and the true distribution.

2. To model the mass of the heaviest halo, we first transform a base distribution using a stack of five spline transformations with 8 knots [5]. The base distribution here is not the standard Gaussian, as is often used with normalizing-flows, but the probability distribution function estimated from the unconditional halo mass function, as described in [20] (see the central histogram in Fig. 1). We found that using this physically motivated base distribution was

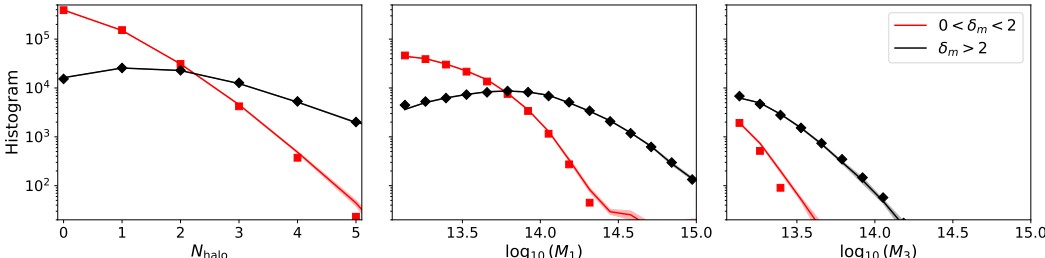

Figure 2: Comparison of mean and standard deviation in the one-point statistics between the true (markers) and CHARM (solid lines) halo catalogs in test simulations. We display the distribution of the total number of halos (left), the mass of the heaviest halo (center), and the mass of the third heaviest halo (right). We present these distributions for sub-selections of voxels with either low (red-colored) or high (black-colored) dark matter density. This test individually compares the performance of each of the three stages of the network (as described in § 4).

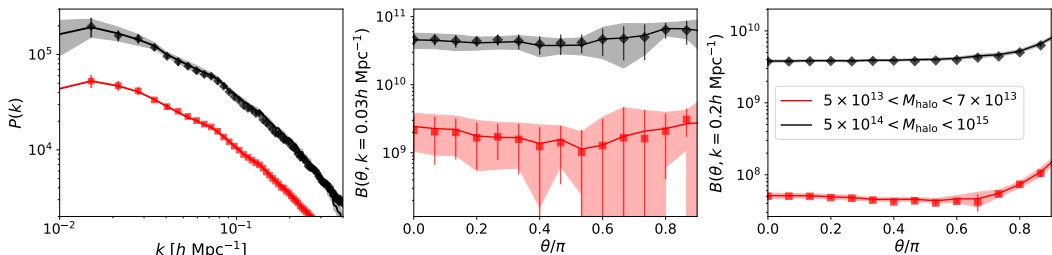

Figure 3: Comparison of mean and standard deviation of power spectrum (left), bispectrum on large scales (center), and bispectrum on small scales (right) statistics between the true (markers) and CHARM (solid lines) halo catalogs in the test simulations. We calculate these statistics for both low-mass (red) and high-mass (black) halos, showing that the CHARM catalog can accurately capture these statistics on both small and large scales. Note that these statisics probe the global distribution of halos over all the voxels and are sensitive to the overall performance of the network.

crucial for obtaining accurate predictions for the heaviest halo. The parameters of the transformations are learned using a separate FCNN. The loss is then calculated as the KL divergence between a known base distribution and this transformed distribution.

3. We learn the distribution of lower halo masses using an auto-regressive neural spline flow [5]. We condition the transformation on the masses of all the heavier halos. Additionally, to ensure a decreasing order of halo masses, for the $j$-th halo in the voxel, we learn the mass difference $M_{j-1} - M_j$ and ensure that this difference is positive. Here, we stack two such spline flows, and their parameters are once again learned using an FCNN. As having more halos in the same voxel becomes a rarer phenomenon, we model the base distribution as proportional to the Gumbel distribution [8], which provides a good initial estimate of this extreme value statistics (see the bottom histogram in Fig. 1).

## 5   Results

To test the performance of our network, we calculate the one-point statistics (histogram), two-point statistics (power spectrum), as well as three-point statistics (bispectrum) from CHARM and true halo catalogs in 10 independent test simulations and compare their means and standard deviation.

In Fig. 2, we compare the one-point statistics of the total number of halos, the mass of the heaviest halo, and the mass of the third heaviest halo. We compare these histograms in both low-density ($0 < \delta_m < 2$, red color) and high-density ($\delta_m > 2$, black color) environments to highlight the differences in these histograms and their dependence on the underlying dark matter density field. We observe a good match for all three histograms in both environmental conditions. That this one-point

comparison individually tests the performance of each of the three parts of the pipeline as mentioned in § 4. We see that for all the cases where the number of halos is significant, our network is accurate at percent level.

In Fig. 3, we compare the mock and true mean power spectrum and bispectrum from the test simulations. To highlight the differences in clustering properties depending on the masses of the halos, we calculate these statistics for a sub-sample of either low-halo masses ($5 \times 10^{13} < M_{\mathrm{halo}}(M_\odot/h) < 7 \times 10^{13}$, red color) or high-mass halos ($5 \times 10^{14} < M_{\mathrm{halo}}(M_\odot/h) < 10^{15}$, black color). On the left plot, we display the power spectrum as a function of scale and find that the statistics match in both large scales and small scales. As the bispectrum is a three-point statistic, it depends on three scales (or two scales and one angle between them). We show the results for the isosceles triangle configuration (hence the two scales are equal) and as a function of the angle between them. We present this for both large-scale and small-scale configurations in the middle and right panels respectively for the two mass selections. We find good agreement between all four cases. Note that two- and three-point statistics jointly test the performance of all the three stages of the network as well as for all the voxels globally.

Moreover, as shown in both Fig. 2 and Fig. 3, we also capture the uncertainity arising due to stochasticity of uncertain initial condtions. This is a natural by-product of the generative model used here which captures the distribution of halo masses and its correlation with underlying matter density. This is another advantage compared to previous studies like [23] which use regression-based models and hence are unable to quantify uncertainities.

We note that estimating the total computational speed-up achieved by using this algorithm, compared to running conventional N-body simulations, is somewhat complicated due to the involvement of different devices. However, running the required PM simulations takes a total of 3 CPU-hours, and training the full network on one V100 GPU requires 2 hours. In contrast, running a high-resolution N-body simulation and identifying halos takes approximately 5000 CPU-hours. This results in a computational speed-up factor of about 1000 when using the CHARM algorithm for halo generation.

# 6    Discussion

In this work, we describe how accelerating cosmological simulations and identifying collapsed dark matter structures (halos) are key to maximizing the information gained from current and upcoming galaxy surveys. We develop a multi-stage generative model that can learn the relationship between the halo distribution and the surrounding dark matter distribution on large scales. We demonstrate how these relationships can be learned using a faster alternative to N-body simulations, which provide correct dark matter densities on large scales. We validate our mock catalogs obtained with CHARM using one-, two-, and three-point statistics under various environmental conditions, demonstrating the accuracy of the method. This results in computational gains of more than a factor of 1000 in generating relevant halo catalogs for galaxy survey analysis.

In this study, we fixed the cosmological parameters and learnt the generative model on a single cosmology. In our next work, we plan to improve the conditional network by accounting for the cosmology dependence before proceeding to data analysis and constraining the cosmological parameters. To push to smaller scales, we will also train the network on higher resolution simulations. Finally, we have currently focused only on generating the halo masses and positions. To model some cosmological observables, halo velocity and concentration are also required. We will develop similar techniques to learn these.

## Acknowledgments and Disclosure of Funding

This work is supported by the Simons Collaboration on "Learning the Universe". The Flatiron Institute is supported by the Simons Foundation.

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
