# OpenReview forum: "CHARM: Creating Halos with Auto-Regressive Multi-stage networks"
_NeurIPS.cc/2023/Workshop/AI4Science — NeurIPS2023-AI4Science Poster_

### Official Review · Reviewer_9WtV · 2023-10-23
**Creating mock halo catalogs by deep neural network**

**Rating:** 6
**Confidence:** 3

**Review:**

A method (CHARM) for creating mock halo catalogs by matching the spatial and mass statistics of halos is proposed in this paper.
One-, two- and three-point statistics are calculated to evaluate the performance.
The results demonstrate the accuracy and speed-up of CHARM.

**Strengths**

1. The research background and data description are clear and detailed.
2. Potential application of this method is promising.

**Weaknesses**

My main concern about this work is the novelty. Compared to references [2, 12, 15, 23], the introduced multistage deep learning method seems not so innovative and attractive to me.

1. Few related works are presented in the paper.
2. Lack of ablation study and comparison with related works and conventional methods.

**Questions**

Besides the weakness, I have some other questions.

1. The ResNet and NSF architectures are not given in the paper, as well as training strategy.
2. It would be better to provide the source code to illustrate the reproducibility.
3. How is 1000x speedup over conventional method derived ?
4. As shown in middle figure of Figure 2, the standard deviation of low $M_{halos}$ is large, I wonder the reason.

---

### Meta-Review · Area_Chair_fLMS · 2023-10-27

**Recommendation:** Accept (Poster)
**Confidence:** 3

**Metareview:**

The authors describe a method to create mock halo catalogs with a machine learning approach, leading to a 1000x speedup over comparable approaches. The application is promising and would lead to useful discussions. The reviewers raise some questions about reproducibility which would be good for the authors to address before final submission.